# Conventional Chromosome Analysis of Fetuses with Central Nervous System Anomalies and Associated Anomalies: Is Anything Changed?

**DOI:** 10.3390/medsci6010010

**Published:** 2018-02-06

**Authors:** Emre Ekmekci, Emine Demirel, Servet Gencdal

**Affiliations:** 1Maternal-Fetal Medicine Unit, Department of Obstetrics and Gynecology, Faculty of Medicine, Izmir Katip Celebi University, Izmir 35330, Turkey; er_em.dr@hotmail.com; 2Obstetrics and Gynecology Clinic, Izmir Ataturk Education and Research Hospital, Ministry of Health, Izmir 35330, Turkey; servetgencdal@hotmail.com

**Keywords:** aneuploidy, central nervous system anomalies, conventional karyotyping, karyotype, prenatal diagnosis

## Abstract

Central nervous system (CNS) abnormalities are often isolated but can accompany various genetic syndromes. In this study, we evaluated conventional karyotype results and associated findings of fetuses that were diagnosed with CNS abnormalities. Cases included in the study were diagnosed with fetal CNS anomalies and underwent conventional karyotyping. Conventional karyotype results of subjects were compared with karyotype results of fetal karyotyped patients as a result of maternal anxiety in a two-year period. In this period, 69 patients were diagnosed with fetal CNS anomalies and 64 of them underwent invasive fetal karyotyping. Of these, 32 patients had isolated CNS anomalies, while 32 were associated with other anomalies. There was no significant difference between karyotype results when compared with the control group (*p* = 0.76). Apart from some specific anomalies, the aneuploidy rate does not significantly differ between fetuses with CNS anomalies and the control group. Advanced genetic evaluation may provide additional diagnostic benefits, especially for this group.

## 1. Introduction

Central nervous system (CNS) anomalies are the most common congenital anomalies. Their prevalence is 0.14–0.16% in live births, although this can be as high as 3–6% in stillbirths [1]. Although the etiology of CNS anomalies is highly heterogeneous and multifactorial, genetic disorders are of great significance to the etiology [2]. There has been a significant drop in neural tube defect cases due to the more widespread use of folic acid in recent years. However, when considering that CNS anomalies develop due to the mutagenic effects of environmental factors, genetic etiology would emerge to be more important [3]. The association of certain CNS anomalies with Trisomy 13 and Trisomy 18 has been proven in various studies [4]. Determining the genetic etiology of such anomalies is important for both counseling about existing pregnancy and recurrence risk in subsequent pregnancies.

Nowadays, the submicroscopic evaluation of chromosomes has gained attention in perinatal medicine. Despite the effectiveness of conventional G-band karyotyping in finding aneuploidies that are major numeric chromosomal disorders and chromosomal deletions larger than 5 Mb (mega-base), this is not efficient for diagnosing minor deletions [5]. Nevertheless, this is still the first choice for genetic testing when an anomaly is detected by ultrasound.

In this study, we evaluated conventional G-band karyotype results of fetuses with CNS anomalies that were diagnosed in a two-year period in our center as well as other system anomalies associated with these karyotypes. Furthermore, we compared karyotype rates with a control group. We aimed to discuss which type of anomalies required more advanced genetic evaluation.

## 2. Materials and Methods

This prospective observational study was conducted in the Department of Obstetrics and Gynecology, Maternal and Fetal Medicine Unit, School of Medicine, Izmir Katip Celebi University, Izmir, Turkey between March 2014 and June 2016. The unit is a tertiary center in the west of Turkey that treats referred patients from the region. The approval for this study was obtained by the clinical board of the department. The study design was in accordance with the Helsinki Declaration and Committee on Publication Ethics Guidelines. The pregnant women diagnosed with fetal CNS anomalies were included in the study as the study group. They were examined in detail for exact diagnoses and possible additional malformations. The diagnoses of anomalies were conducted by a Voluson E6 Expert ultrasonic diagnostic system (GE Healthcare, Little Chalfont, UK). All patients were treated by medical genetics specialists. According to the opinion of medical genetics specialists, the option for invasive karyotyping was provided to all patients and they were informed about the benefits and risks of these invasive test procedures. Informed consent was taken from all patients just before invasive test procedures. According to gestational age, chorion villus sampling, amniocentesis, and cordocentesis were performed. All of the karyotypes were obtained prenatally and culture failures were excluded from the study. Other system anomalies were also recorded. The control group included patients who had undergone invasive fetal karyotyping due to maternal anxiety. Maternal anxiety allows for prenatal screening tests without high risks of aneuploidies. In cases with no major findings that are indicative for karyotyping, no findings detected on prenatal sonography and/or without an abnormal karyotyped pregnancy history, a patient can still opt for an invasive test at her request. Obtained karyotype results were compared between two groups. The Student’s *t*-test was used to compare abnormal karyotype rates between groups. Statistical analysis was performed with MedCalc Statistical Software version 17.9.7 (Med Calc Software, Ostend, Belgium).

## 3. Results

In this period, a total of 1649 patients was referred to our center and 69 CNS anomalies were detected (4.1%). The mean age for the subject group was 26.2 ± 4.2 years, and it was 34 ± 3.5 years for the control group. The median gravida was two, while the parity was two in the subject group and three in the latter group. Five patients rejected karyotyping and 64 invasive procedures were performed for karyotyping. Thirty-two of these were amniocentesis, 17 were cordocentesis, and 15 were chorionic villous sampling. The median gestational age at diagnosis was 18 weeks. The detected CNS and associated anomalies are listed in Table 1.

In this period of time, a total of 134 invasive procedures was performed due to maternal anxiety and six abnormal karyotypes were detected in this group. The median gestational age at diagnosis was 17 weeks. Forty-three pregnancies underwent induced abortion due to parents’ request, and diagnosed anomalies were confirmed postnatally by macroscopic tests or autopsy. The abnormal karyotypes in the two groups are defined in Table 2. There was no statistically significant difference between the two groups (*p* = 0.76).

## 4. Discussion

In this study, our objective was to estimate the prevalence of abnormal karyotypes with conventional G-band karyotyping in fetuses with CNS anomalies and compare the results with the control group. We aimed to evaluate the adequacy of genetic counseling with only conventional G-band karyotyping for parents having pregnancies with CNS anomalies.

In total, 69 CNS anomalies were detected in 1649 pregnancies, and this ratio (4.1%) was high when compared to the general population. Our center is a referral center and referred patients apply from various centers, which seems to be causing this high rate. The CNS anomaly rate was reported as 0.28% in a study conducted in Turkey [6].

Spina bifida, exencephaly, and encephalocele, which are named as neural tube defects, are the most common and serious CNS anomalies. Their incidence decreases with the replacement of folic acid, which shows the effect of environmental factors [7]. When considering that all Arnold Chiari type-2 cases were secondary to open spina bifida, exencephaly, open spina bifida, and encephalocele cases comprise 60.8% (42/69) of all CNS anomalies, while 59.5% of this group was isolated.

Although isolated ventriculomegaly is a sonographic finding rather than an anomaly, our rate of this finding in all CNS anomalies was 5.7%, which seems to be quite low when compared to the studies that included isolated ventriculomegaly. This rate is reported as 52.8% by Amer et al. [8]. Our lower rate may be due to isolated ventriculomegaly cases not developing as isolated cases and thus, not being grouped as isolated. Pilu et al., reported the aneuploidy rate as 3.8% for isolated ventriculomegaly [9]. In this study, four isolated ventriculomegaly cases had normal karyotypes.

Dandy–Walker malformation was reported to have a 50% association with aneuploidies when associated with other anomalies. However, this is frequently sporadic when isolated [10]. In our four Dandy-Walker malformation (DWM) cases, we detected one Trisomy 18 and all cases were non-isolated.

Although isolated holoprosencephaly cases are commonly sporadic and have normal karyotypes, 25–50% have aneuploidy when associated with other anomalies. Even 70% of Trisomy 13 cases have holoprosencephaly [11]. In our study group, one Trisomy 13 and one triploidy patient were diagnosed with holoprosencephaly, which presented in a total of five cases (40%). Apart from one lobar holoprosencephaly case, all cases were associated with other anomalies.

Although various rates have been reported in the literature and the aneuploidy rate is quite low for neural tube defects, Trisomy 13 and 18 are the most common aneuploidies. Ceylaner et al. reported a rate of 2.8% for aneuploidies [12]. Ekin et al. found a chromosomal abnormality rate of 1.8% in fetuses with open neural tube defects [13].

Stoll et al. reported an aneuploidy rate of 2.5% for 441 neural tube defect cases in their study [14]. Interestingly, there is a remarkable association of deletions on the long arm of the 13th chromosome with various anomalies, especially CNS anomalies. In a study conducted in Italy, the authors reported eight CNS anomalies, six eye anomalies, nine facial dysmorphism cases, and 10 extremity anomalies from 14 deletions of 13q cases [15]. In our study, one inversion of a 13q patient had open spina bifida associated with ductus venosus agenesis and ventricular septal defect (VSD). Ballarati et al. reported that loss of function in *ZIC2* and *ZIC5* genes located on the long arm of the 13th chromosome results in neural tube defects or DWM according to the degree of functional loss in their study using the micro-array technique [15]. The inversion we detected between the short and long arms of the 13th chromosome, which is large enough to be diagnosed by microscopic evaluation only, seems to support the relationship between structural abnormalities in the long arm of the 13th chromosome and neural tube defects. We need more sensitive techniques, such as the micro-array, to identify submicroscopic deletions.

We compared the rates of abnormal karyotypes in the CNS anomalies group with the control group and found no statistically significant difference (*p* = 0.76). However, it is important to note that the mean age of our maternal anxiety group was older and the abnormal karyotype rate was higher than the general population. Some study limitations should be acknowledged, including the most important ones of sample size, distribution of CNS anomalies, and accretion at neural tube defects. Despite these limitations, our results confirm the previous knowledge about this topic and highlight the requirement for more a detailed genetic evaluation of CNS anomalies.

Conventional karyotyping alone seems insufficient to provide enough genetic counseling in pregnancies with CNS anomalies. With the application of more sensitive and advanced genetic analyses, the role of genetics in CNS anomalies will be fully revealed. Nowadays, chromosomal micro-arrays seem to be promising.

## Figures and Tables

**Table 1 medsci-06-00010-t001:** Diagnosed central nervous system anomalies, karyotypes, and associated anomalies.

CNS Anomaly	Abnormal Karyotype	Associated Anomaly	
Arnold-Chiari Type 2 (*n* = 17)	1/17 (5.8%)	1	Outlet VSD
		2	Muscular VSD
		3	Perimembranous VSD, ductus venosus agenesis **46, inv (13) (p1q13)**
		4	AVSD
		5	Rocker-bottom feet
		6	Club feet
		7	Arachnoid cyst
Exencephaly (*n* = 15)	-	1	AVSD
		2	AVSD
		3	AVSD
		4	Right isomerism
		5	Omphalocele, ectropia cordis
Encephalocele (*n* = 10)	-	1	AVSD
		2	AVSD
		3	AVSD, DWM
		4	Club foot
		5	Multicystic kidneys, polydactyly (**Meckel–Gruber**)
Holoprosencephaly (*n* = 5)	2/5 (40%)	1	Rocker-bottom feet, clenched hand, cleft lip-palate, hypotelorism, proboscis, hypoplastic left heart **Trisomy 13**
		2	Median cleft lip, rocker-bottom feet **Triploidy**
		3	Median cleft lip, rocker-bottom feet.
		4	Hypotelorism, cleft lip-palate
Isolated ventriculomegaly (*n* = 4) (AW: 10–14 mm)	-		-
Dandy–Walker Malformation (*n* = 4)	1/4 (25%)	1	AVSD, rocker-bottom feet, and clenched hand **Trisomy 18**
		2	Omphalocele, tetralogy of Fallot
		3	Club feet
		4	Hydrops, agenesis of corpus callosum, aortic coarctation
Agenesis of corpus callosum (*n* = 5)	-	1	Double outlet right ventricle
Aqueductal stenosis (*n* = 3)	-	1	Aortic coarctation
Arachnoid cyst (*n* = 2)	-	1	Spina bifida
Subepandimal periventricular heterotopia (*n* = 1)	-	-	-

**Abbreviations**: central nervous system (CNS); ventricular septal defect (VSD); atrioventricular septal defect (AVSD); Dandy-Walker malformation (DWM); lateral ventricular atrial width (AW).

**Table 2 medsci-06-00010-t002:** Distribution of abnormal karyotypes between the study and control groups.

	Karyotype	Trisomy 21	Trisomy 18	Trisomy 13 (47, +13)	46, Inv (13) (p1q13)	Klinefelter (47, XXY)	Triploidy (69, XXX)
Indications							
**Maternal Anxiety** ***n*: 134, 4.4%**		4	1	-	-	1	-
**CNS Anomalies** ***n*: 64, 6.2%**		-	1	1	1	-	1

**Abbreviations**: central nervous system (CNS).

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
