# Peer review of "Conventional Chromosome Analysis of Fetuses with Central Nervous System Anomalies and Associated Anomalies: Is Anything Changed?"

_medsci, 2018, doi:10.3390/medsci6010010_

Round 1
Reviewer 1 Report
Dear Editor,
Thank you for asking me to review the manuscript titled “Conventional chromosome analysis of fetuses with central nervous system anomalies and associated anomalies; is anything changed?” submitted by Ekmekci et al. for consideration for publication.
I have following comments hoping that they will improve the manuscript further:
1. Line 29: I am not sure what “0,14-0,” means. Please clarify.
2. Line 34: please spell out “Tri” as “trisomy”
3. Please add number of patients referred your center during the study period in Materials and Methods section.
4. Please define study groups clearly.
5. Please add statistical method(s) used for comparison.
6. In results section
a. Please add a table for patient demographics.
b. Title of Table 2 appears to be the same as Table 1. Please clarify.
c. In table 1, please consider adding associated anomalies into the table.
7. The manuscript should be reviewed by a native English speaker for grammar and wording.
Author Response
Thanks to reviewer for recommendations and comments. We have made gross changes in manuscript according to your comments.
We prepared the paper with required revisions. Definition of research design is prepared more accurately. Methods are described more clearly. Results are defined in tables according to your comments and we have added associated anomalies in table according to your recommendations.
1. The prevalance numbers 0,14 - 0,16% is corrected.
2. Tri is corrected as Trisomy in line 34
3. Study groups are defined clearly.
4. Number of patients referred to ur clinic is specified in results.
5. Statistical method is added
6. Title of Table 2 is corrected.
7. Linguistic revision and design of paper is is done by journal's editting service.
Reviewer 2 Report
You made all the correction requested and the results and the objective of the project are clear. The message is not original. Nowadays, you already know that the standard karyotype is not enough to identify genomic rearrangements associated with fetal malformations (Schaffer LG 2012). However, since cases included in the study only concern CNS abnormalities, the paper is worth reporting.
Reviewer 3 Report
The number of fetuses with CNS defects is small with high proportion of Neural Tube Defects with relatively low rate of aneuploidies. The control group is formed by advanced maternal age women at increased chromosomal risk so no statistical difference between the two groups is emerging. It should be interesting to know if the isolated anomalies and karyotype were confirmed after birth/termination. I agree with the authors that standard karyotype is now absolutely insufficient to identify a genetic etiology underlying fetal brain anomalies and to ensure a specific genetic counselling.
Author Response
Thanks to reviewer for recommendations and comments. We have made gross changes in manuscript according to reviewers' comments.
We prepared the paper with required revisions. Definition of research design is prepared more accurately. Methods are described more clearly. Results are defined in tables and we have added associated anomalies in table.
Paper design and linguistic revision is made by journal's editing service.
Yes I agree with you about this topic isnot a new insight in medicine. We wanted to report these results one more time before the new technology micro-array is more and more having place in our practice nowadays. And all studies are going towards to microarray.
We used control group as maternal anxiety indicated karyotyping because this group is not a high risk group but it is also noted and indicated in manuscript that this group has advanced maternal age compared to study group.
Reviewer 4 Report
The manuscript does not add any significant kwnoledge about the issue presented. Classification o congenital malformations is inadequate and data about patients is unsufficient. Were the associated anomalies detected postnatally or by sonogram only? cases with specific diagnosis (i.e Meckel-Gruber) should be removed from the group of patients since there is no evidence that fetal karyotype could be of any help. Conclusions have no clinical interest
Author Response
Thanks to reviewer for recommendations and comments. We have made gross changes in manuscript according to reviewers' comments.
We prepared the paper with required revisions. Definition of research design is prepared more accurately. Methods are described more clearly. Results are defined in tables and we have added associated anomalies in table.
Paper design and linguistic revision is made by journal's editing service.
'The median gestational age at diagnosis was 17 weeks. 43 pregnancies have undergone induced abortion due to parents' request and diagnosed anomalies are postnatal confirmed macroscopic or by autopsy.' This data is added to results.
Yes I agree with you about this topic isnot a new insight in medicine. We wanted to report these results one more time before the new technology micro-array is more and more having place in our practice nowadays. And all studies are going towards to microarray.
We did not exclude the one patient compatible with Meckel- Gruber , because specific mutation could not have been studied.